# Efficacy and Safety of SGLT-2 Inhibitors for Treatment of Diabetes Mellitus among Kidney Transplant Patients: A Systematic Review and Meta-Analysis

**DOI:** 10.3390/medsci8040047

**Published:** 2020-11-17

**Authors:** Api Chewcharat, Narut Prasitlumkum, Charat Thongprayoon, Tarun Bathini, Juan Medaura, Saraschandra Vallabhajosyula, Wisit Cheungpasitporn

**Affiliations:** 1Department of Medicine, Mount Auburn Hospital, Harvard Medical School, Cambridge, MA 02138, USA; 2Department of Medicine, Division of Nephrology and Hypertension, Mayo Clinic, Rochester, MN 55905, USA; 3Department of Medicine, University of Hawaii, Honolulu, HI 96822, USA; narutprasitlumkum@gmail.com; 4Department of Internal Medicine, University of Arizona, Tuscon, AZ 85721, USA; tarunjacobb@gmail.com; 5Department of Internal Medicine, Division of Nephrology, University of Mississippi Medical Center, Jackson, MS 39216, USA; jmedaura@umc.edu; 6Section of Interventional Cardiology, Department of Medicine, Division of Cardiovascular Medicine, Emory University School of Medicine, Atlanta, GA 30322, USA; saraschandra.vallabhajosyula@emory.edu

**Keywords:** SGLT-2 inhibitors, kidney transplant, renal transplant, transplantation, meta-analysis

## Abstract

Background: The objective of this systematic review was to evaluate the efficacy and safety profiles of sodium-glucose co-transporter 2 (SGLT-2) inhibitors for treatment of diabetes mellitus (DM) among kidney transplant patients. Methods: We conducted electronic searches in Medline, Embase, Scopus, and Cochrane databases from inception through April 2020 to identify studies that investigated the efficacy and safety of SGLT-2 inhibitors in kidney transplant patients with DM. Study results were pooled and analyzed utilizing random-effects model. Results: Eight studies with 132 patients (baseline estimated glomerular filtration rate (eGFR) of 64.5 ± 19.9 mL/min/1.73 m^2^) treated with SGLT-2 inhibitors were included in our meta-analysis. SGLT-2 inhibitors demonstrated significantly lower hemoglobin A1c (HbA1c) (WMD = −0.56% [95%CI: −0.97, −0.16]; *p* = 0.007) and body weight (WMD = −2.16 kg [95%CI: −3.08, −1.24]; *p* < 0.001) at end of study compared to baseline level. There were no significant changes in eGFR, serum creatinine, urine protein creatinine ratio, and blood pressure. By subgroup analysis, empagliflozin demonstrated a significant reduction in body mass index (BMI) and body weight. Canagliflozin revealed a significant decrease in HbA1C and systolic blood pressure. In terms of safety profiles, fourteen patients had urinary tract infection. Only one had genital mycosis, one had acute kidney injury, and one had cellulitis. There were no reported cases of euglycemic ketoacidosis or acute rejection during the treatment. Conclusion: Among kidney transplant patients with excellent kidney function, SGLT-2 inhibitors for treatment of DM are effective in lowering HbA1C, reducing body weight, and preserving kidney function without reporting of serious adverse events, including euglycemic ketoacidosis and acute rejection.

## 1. Introduction

Diabetes mellitus (DM) is the leading cause of end-stage kidney disease (ESKD) worldwide [1]. In the United States, Organ Procurement Transplant Network/Statistics on Donation and Transplantation in the United States (OPTN/SRTR) reported that nearly 40% of patients on transplant waiting list in 2018 had DM with an ongoing upward trend [2]. In addition, approximately 15–30% of nondiabetic kidney transplant recipients develop new-onset diabetes after transplant [3,4,5], resulting in a high prevalence of kidney transplant recipients with preexisting and post-transplant diabetes mellitus (PTDM), ranging up to 74% depending on country, ethnicity, and criteria of diagnosis [6,7]. DM among kidney transplant recipients has been associated with higher rates of cardiovascular disease, infectious complications, graft loss, and mortality [8,9,10,11]. Hence, controlling blood sugar is necessary in order to prevent these poor outcomes among kidney transplant recipients [12]. Lifestyle modifications, including dietary modification, weight reduction, and exercise, along with pharmacologic treatment, are required for treatment of DM among kidney transplant recipients [13,14].

Sodium-glucose co-transporter 2 (SGLT-2) inhibitors are novel oral glucose-lowering medications that inhibit glucose reabsorption in the proximal tubule and promote renal excretion of glucose [15]. Glycemic efficacy of SGLT-2 inhibitors is considered to be relatively weak compared to other glucose-lowering medications with 0.4% to 1.1% reduction in hemoglobin A1C (HbA1c) levels [16,17,18]. This glucose-lowering effect is independent of beta-cell function and insulin sensitivity [19]. Thereby, SGLT-2 inhibitors do not generally cause hypoglycemia [20,21,22,23]. Recent meta-analyses demonstrated the cardio- and renal protective effects such as reduce the progression of diabetic nephropathy, diminish cardiovascular mortality, lower rates of hospitalized heart failure, and decrease weight among patients with DM [24,25,26]. Nevertheless, some adverse effects of SGLT-2 inhibitors have been reported, such as urinary tract infection, genital mycosis, acute kidney injury (AKI), hypotension, bone fracture, diabetic ketoacidosis, and amputation [27,28,29,30]. However, most of the studies that reported these efficacious and adverse effects focused on only the non-kidney transplant population [31,32,33,34,35,36,37,38,39,40,41,42,43,44,45,46]. The efficacy and safety data for the use of SGLT-2 inhibitors in kidney transplant recipients with DM remain unclear.

Therefore, we conducted the present systematic review and meta-analysis to assess the efficacy and safety of SGLT-2 inhibitors for treatment of DM among kidney transplant recipients.

## 2. Materials and Methods

### 2.1. Data Sources and Search Strategies

A comprehensive search of several databases from each database’s inception to 30 April 2020, any language, was conducted. The databases included Ovid MEDLINE(R) and Epub Ahead of Print, In-Process and Other Non-Indexed Citations, Daily, Ovid EMBASE, Ovid Cochrane Central Register of Controlled Trials, Ovid Cochrane Database of Systematic Reviews, and Scopus. The search strategy was designed and conducted by an experienced librarian with input from the study’s principal investigator. Controlled vocabulary supplemented with keywords was used to search for studies of SGLT-2 inhibitors in kidney transplant patients. The actual strategy listing all search terms used is available in the Appendix A. This study was conducted by the Preferred Reporting Items for Systematic Reviews and Meta-Analysis (PRISMA) statement [47].

### 2.2. Study Selection

Given limited data, our study included any studies, including case series and randomized controlled trials, examining the efficacy and safety of any SGLT-2 inhibitors among kidney transplant patients. We excluded case reports or animal studies. Case series were defined as a study that included multiple cases receiving SGLT-2 inhibitors without a control group. For cohort studies or randomized controlled trials, we focused only on treatment group with SGLT-2 inhibitors in order to be comparable and consistent with case series. The quality assessment tool for case series studies proposed by the National Heart, Lung, and Blood Institute (NHLBI) was used to rate the quality [48]. Studies with a score of 0–3, 4–6, and ≥7 were considered as low, fair, and good quality, respectively. There were no restrictions on language, sample size, or study duration.

Retrieved articles were individually reviewed for eligibility by two investigators (A.C. and N.P.) utilizing a systematic review management program called Covidence, which has been selected by Cochrane to become a standard production platform for their reviews. Disagreements were addressed and resolved by third investigator (W.C.).

### 2.3. Data Extraction

The following data were extracted: first author name, year of publication, number of participants in SGLT-2 inhibitors group, duration of follow-up, types of SGLT-2 inhibitors, mean age, sex, and duration of transplantation. Efficacy outcomes included estimated glomerular filtration rate (eGFR), serum creatinine, urine protein-creatinine ratio, HbA1C, systolic blood pressure (SBP), diastolic blood pressure (DBP), body mass index (BMI), and body weight. Safety profile outcomes included incidence of urinary tract infection, genital mycosis, euglycemic ketoacidosis, acute kidney injury, acute rejection, ulcer, and cellulitis.

### 2.4. Data Synthesis and Statistical Analysis

A random-effects model was used due to the expected clinical heterogeneity in the included populations. We calculated weight mean difference (WMD) for the difference in mean value between baseline and at the end of study for continuous variables among SGLT-2 inhibitors group. All pooled estimates were shown with 95% confidence intervals (CIs). Heterogeneity among effect sizes estimated by individual studies was described with the I^2^ statistic and the chi-square test. A value of I^2^ of 0% to 25% represents insignificant heterogeneity, 26% to 50% low heterogeneity, 51% to 75% moderate heterogeneity, and 76 to 100% high heterogeneity [49].

Publication bias was evaluated using funnel plots, and the Egger test was used to assess the asymmetry of the funnel plot. A *p*-value of less than 0.05 indicates the presence of publication bias. The meta-analysis was performed by STATA/IC 14.1 (StataCorp LLC, Lakeway, TX, USA).

## 3. Results

### 3.1. Characteristics and Quality of Studies

A total of 119 potentially relevant articles were identified and screened. Fourteen articles were assessed in detail, of which 8 studies with 132 participants fulfilled the eligibility criteria and were included in our meta-analysis (Figure 1).

Characteristics of each study were shown in Table 1. One study was randomized controlled trial [43], while the other seven studies were categorized as case series [50,51,52,53,54,55]. Given these different study designs, we extracted data from only treatment group for the randomized controlled trial and compared each parameter level between at baseline and the end of study. Sample sizes of the treatment group among 8 studies ranged from 8 to 22 participants. The mean age of patients ranged from 46 to 66 years old. Time after kidney transplantation varied from 3 to 20 years. Baseline eGFR among all included patients was 64.5 +/− 19.9 mL/min/1.73 m^2^.

In terms of the quality assessment, seven studies [43,50,51,52,53,54,55] were graded as good and only one study [56] was graded as fair. There were no poor-quality studies.

### 3.2. Efficacy of SGLT-2 Inhibitors on Kidney Function

Seven studies with 124 patients treated with SGLT-2 inhibitors did not show a significant decline in eGFR between baseline and the end of the study (WMD = −2.51 mL/min/1.73 m^2^ [95%CI: −5.03, 0.02]; *p* = 0.06, I^2^ = 0%) (Figure 2).

Five studies with 58 participants treated with SGLT-2 inhibitors demonstrated no significant difference in serum creatinine between levels at baseline and at the end of study (WMD = −0.05 mg/dl [95%CI: −0.12, 0.03]; *p* = 0.21, I^2^ = 0%). Three studies with 38 participants treated with SGLT-2 inhibitors reported no significant difference in urine protein creatinine ratio between levels at baseline and end of study (WMD = −211 mg/g [95%CI: −655, 232]; *p* = 0.35, I^2^ = 93.2%) (Table 2).

There was no significant decline in eGFR comparing between levels at baseline and at either 6 months among 5 studies with 72 participants (WMD = −2.38 mL/min/1.73 m^2^ [95% CI: −5.13, 0.37]; *p* = 0.09, I^2^ = 0%) or between levels at baseline and at 12 months among 4 studies with 68 participants (WMD = −0.35 mL/min/1.73 m^2^ [95% CI: −5.66, 4.97]; *p* = 0.90, I^2^ = 0%). In terms of urine protein creatinine ratio, there was no significant difference between levels at baseline and at 6 months (WMD = −211 mg/g [95% CI: −655, 232]; *p* = 0.35, I^2^ = 93.2%). For serum creatinine, there was no significant difference between levels at baseline and at 12 months (WMD = −0.05 mg/dl [95% CI: −0.15, 0.05]; *p* = 0.32, I^2^ = 0%) (Table 3).

### 3.3. Efficacy of SGLT-2 Inhibitors on Glycated Hemoglobin

Eight studies with 132 participants treated with SGLT-2 inhibitors demonstrated significantly lower HbA1C at the end of study compared to baseline (WMD = −0.57% [95% CI: −0.97, −0.16]; *p* = 0.006, I^2^ = 85.2%) (Figure 3 and Table 2).

Five studies with 76 participants treated with SGLT-2 inhibitors revealed a significant lower in HbA1C at 12 months compared to baseline (WMD = −0.58% [95% CI: −1.12, −0.05]; *p* = 0.03, I^2^ = 79.8%). However, comparing HbA1C levels at 6 months and baseline, 5 studies with 72 participants showed no significant difference (WMD = −0.22% [95% CI: −0.62, 0.18]; *p* = 0.28, I^2^ = 41.6%) (Table 3).

### 3.4. Efficacy of SGLT-2 Inhibitors on Blood Pressure and Body Weight

Six studies with 82 participants treated with SGLT-2 inhibitors failed to demonstrate significant decline in SBP (WMD = −3.24 mmHg [95% CI: −7.92, 1.45]; *p* = 0.18, I^2^ = 21.3%) and DBP (WMD = −1.49 mmHg [95% CI: −3.81, 0.83]; *p* = 0.21, I^2^ = 0%). Eight studies with 132 participants treated with SGLT-2 inhibitors demonstrated a significant decrease in body weight between baseline and the end of study (WMD = −2.15 kg [95% CI: −3.07, −1.23]; *p* < 0.001, I^2^ = 0%). However, there was no significant change in BMI between at baseline and end of study among 3 studies with 38 participants treated with SGLT-2 inhibitors (WMD = −1.20 kg/m^2^ [95% CI: −2.67, 0.27]; *p* = 0.11, I^2^ = 21.4%) (Table 2).

At 6 months, participants treated with SGLT-2 inhibitors had lower BMI (WMD = −0.80 kg/m^2^ [95% CI: −1.38, −0.22]; *p* = 0.007, I^2^ = 0%) and body weight (WMD = −2.49 kg [95% CI: −4.15, −0.84]; *p* = 0.003, I^2^ = 0%) than baseline, while at 12 months, only body weight was significantly lower than baseline (WMD = −1.97 kg [95% CI: −3.21, −0.73]; *p* = 0.002, I^2^ = 0%) but not BMI (WMD = −2.70 kg/m^2^ [95% CI: −6.03, 0.61]; *p* = 0.11, I^2^ = 16.1%) (Table 3).

### 3.5. Safety Profiles of SGLT-2 Inhibitors

In total of 8 studies, 14 out of 132 participants had urinary tract infection. One participant out of 72 participants from 5 studies had genital mycosis. No euglycemic ketoacidosis were reported in four studies. Based on 4 studies, 1 out of 28 participants had acute kidney injury. One study reported one small ulcer in lower extremity and another study reported one case of cellulitis. Based on five reported studies, no rejections were found during follow-up period (Table 4).

### 3.6. Subgroup Analysis

In the subgroup analysis for type of SGLT-2 inhibitors, 2 studies with 30 participants treated with empagliflozin demonstrated a lower BMI at end of study compared to baseline (WMD = −0.82 kg/m^2^ [95% CI: −1.41, −0.24]; *p* = 0.006, I^2^ = 0%). Four studies with 48 participants treated with empagliflozin showed a lower body weight at end of study compared to baseline (WMD = −2.17 kg [95% CI: −3.20, −1.15]; *p* < 0.001, I^2^ = 0%). There were no significant changes in kidney function, HbA1C, SBP, and DBP between baseline and at end of study among participants treated with empagliflozin. On the other hand, 2 studies with 34 participants treated with canagliflozin reported a significant decline in HbA1C (WMD = −0.87% [95% CI: −1.46, −0.27]; *p* = 0.004, I^2^ = 0%) and SBP (WMD = −7.15 mmHg [95% CI: −14.27, -0.03]; *p* = 0.04, I^2^ = 0%). There were no significant changes in kidney functions, BMI, and body weight between baseline and at end of study among participants treated with canagliflozin (Table 3).

### 3.7. Assessment of Publication bias

The Egger’s test for our main primary outcome of eGFR change between levels at baseline and end of study was not significant (*p* = 0.08). Moreover, a funnel plot for eGFR of the included studies did not suggest any asymmetry. Therefore, publication bias was less likely to occur in our meta-analysis (Figure 4).

## 4. Discussion

Our study is the first meta-analysis to demonstrate the efficacy and safety of SGLT-2 inhibitors in treating kidney transplant recipients with DM. SGLT-2 inhibitors effectively lowered HbA1C when treated for at least 12 months and reduced body weight when treated for at least 6 months. Moreover, eGFR, serum creatinine levels, and urine protein-creatinine ratio among patients treated with SGLT-2 inhibitors were stable throughout the follow-up period. Empagliflozin revealed efficacy in reducing body weight, while canagliflozin showed efficacy in diminishing HbA1C and SBP. Reported adverse effects of SGLT-2 inhibitors included urinary tract infection (43.8%), small ulcers in lower extremities (10%), cellulitis (10%), AKI (3.6%), and genital fungal infection (1.4%), respectively. No euglycemic ketoacidosis and acute rejection events were reported.

SGLT-2 inhibitors inhibit sodium-glucose reabsorption at proximal renal tubules, causing osmotic diuresis and natriuresis [15]. As a result, the glucose and HbA1C levels drop [15,26]. However, this glycemic efficacy was modest and limited by the filtered load of glucose and the osmotic diuresis [16,17]. A recent meta-analysis of SGLT-2 inhibitors among diabetic non-kidney transplant patients with chronic kidney disease demonstrated a modest reduction in HbA1C, body weight, and albuminuria [57]. Our findings among kidney transplant recipients suggested similar effects except for albuminuria. The modest effect in lowering HbA1C of around 0.6% in our kidney transplant recipients with DM was consistent with previously reported mean reduction ranging between 0.4 and 1.1% among general diabetic patients [16,17]. In light of body weight, previous literature reported a weight loss of 2 to 3 kg among diabetic patients treated with SGLT-2 inhibitors [17,58]. This finding aligns with our result of 2 kg weight reduction. Although we pooled data mostly from case series, our findings suggested similar efficacy of SGLT-2 inhibitors between kidney transplant recipients with DM and the non-kidney transplant diabetic population [17,58]. In terms of preserving kidney functions, our study did not show a significant change of eGFR, serum creatinine, and urine protein. Even though it is possible that the follow-up times (6 to 12 months) were not long enough to observe these differences, at least during these periods of follow-up, there was no worsening kidney function among kidney transplant recipients with DM treated with SGLT-2. Previous major clinical trials of empagliflozin (EMPA-REG OUTCOME) and canagliflozin (CANVAS Program) suggested a benefit of SGLT-2 inhibitors in reducing progression of proteinuria [40,59]. However, due to limited data, we did not have control groups and thus we could not conclude whether treated with SGLT-2 inhibitors helps slow the progression of proteinuria among kidney transplant recipients with DM.

In terms of blood pressure, theoretically, a decrease in body weight combined with osmotic diuresis and reduction of total body sodium can result in lowering blood pressure around 4–6 mmHg in SBP and 1–2 mmHg in DBP, as shown in non-kidney transplant patients with DM [60]. However, our study demonstrated no significant reduction in blood pressure among kidney transplant recipients treated with SGLT-2 inhibitors. This could be explained by more complicated pathogenic mechanisms of hypertension among kidney transplant recipients, including but not limited to side effects from immunosuppressive regimens and the presence of native kidneys [43,61]. Another possible explanation is the insufficient sample size to detect these differences. Although we observed the reduction of SBP among those treated with canagliflozin but not empagliflozin, these incongruent effects might be explained by different baseline characteristics such as higher eGFR and shorter transplant duration, which might reflect a better graft function among canagliflozin group compared to empagliflozin.

In addition to the effect on diminishing intraglomerular hyperfiltration and hypertension, SGLT-2 inhibitors also demonstrate anti-inflammatory, antifibrotic, and protective effects against deregulation of extracellular matrix evidenced by a reduction in serum levels of TNF receptor 1, IL-6, matrix metalloproteinase 7, and fibronectin 1 [62]. Moreover, SGLT-2 inhibitors also reduce serum level of leptin, c-reactive protein, and IL-1β secretion via ROS-NLRP3-caspase-1 pathway [63,64]. Furthermore, SGLT-2 inhibitors also showed antioxidative effects via activation of SIRT1/AMPK, suppression of Akt/mTOR signaling pathway [65]. Some studies also found a decreased level in myeloperoxidase among SGLT-2 inhibitors group, suggesting a lower oxidative damage to vascular endothelium [66]. These potential effects of SGLT-2 inhibitors might benefit kidney transplant recipients in which inflammation and fibrosis are the major process in allograft rejection.

In light of safety profiles of SGLT-2 inhibitors among kidney transplant recipients with DM, the incidence of urinary tract infection was 43.8% in our study vs. 38.0% among general kidney transplant recipients [67], which was not significantly different (*p* = 0.13). This was supported by previous meta-analysis of non-kidney transplant diabetic population, which revealed an increased risk of genital infection but found no significant increased risk of urinary tract infection [68]. In our study, we found 1.4% of genital mycosis. However, due to limited data, we did not have a control group or incidence reported among general kidney transplant recipients to compare. SGLT-2 inhibitors have been found to be associated with a higher risk of euglycemic ketoacidosis via the mechanism involving the decrease in insulin and increase glucagon secretion, which stimulates a shift of glucose to fat metabolism and promotes ketogenesis [29,69]. Nevertheless, studies included in our meta-analysis did not report any euglycemic ketoacidosis events. However, it is possible that sample size or follow-up time was not adequate to capture this event. Slight increase in serum creatinine with early reductions in eGFR (of ~4 mL/min/1.73 m^2^) during the first ~3–6 weeks of treatment SGLT-2 inhibitors has been observed among non-kidney transplant patients, and data has supported this [26,70]. Lowering serum creatinine and increasing eGFR after discontinuation of SGLT-2 inhibitors suggest functional nature of the GFR reduction induced by these drugs [26,70]. Nevertheless, a case of biopsy-proven osmotic nephrosis in a non-kidney transplant patient with DM treated with canagliflozin was recently reported [71]. While our study demonstrated the incidence of AKI of 3.6% among kidney transplant recipients treated with SGLT-2 inhibitors, data on allograft kidney biopsy are limited and require further study.

There are some limitations in our meta-analysis. Firstly, included studies were mostly case series without a control group, and the sample size was fairly small. Secondly, the median time of follow-up was 12 months. Hence, some complications might not be have been observed during 1–2 years of follow-up. Moreover, we did not have a control group to compare to, so given the design of each study and insufficient data from each study, we could not adjust for potential confounders such as concomitant antidiabetic medications and duration of diabetes. Furthermore, the data on cardiovascular outcomes were lacking. Therefore, large randomized controlled trials with long-term follow-up are required to investigate the effects of SGLT-2 inhibitors on outcomes after kidney transplantation, including cardiovascular events, delayed graft rejection and mortality risk, as well as safety profiles such as hypovolemia [72]. Conversely, several strengths should be highlighted. First, subgroup analyses allowed us to investigate the effect of SGLT-2 inhibitors based upon type of SGLT-2 inhibitors and change of each parameter at 6 and 12 months. Additionally, we performed random-effects model and searched for heterogeneity by incorporating additional sources of variability between studies to make our results more robust given possible different backgrounds of population in each study.

## 5. Conclusions

In conclusion, our study suggested that SGLT-2 inhibitors were effective in lowering HbA1C when treated at least 12 months, reducing body weight when treated at least 6 months, and preserving kidney function among kidney transplant recipients with DM without reported serious adverse events, including euglycemic ketoacidosis and acute rejection. Based on our findings, future randomized controlled trials are required to further investigate the efficacy and safety of SGLT-2 inhibitors, including long-term allograft and patient survival.

## Figures and Tables

**Figure 1 medsci-08-00047-f001:**
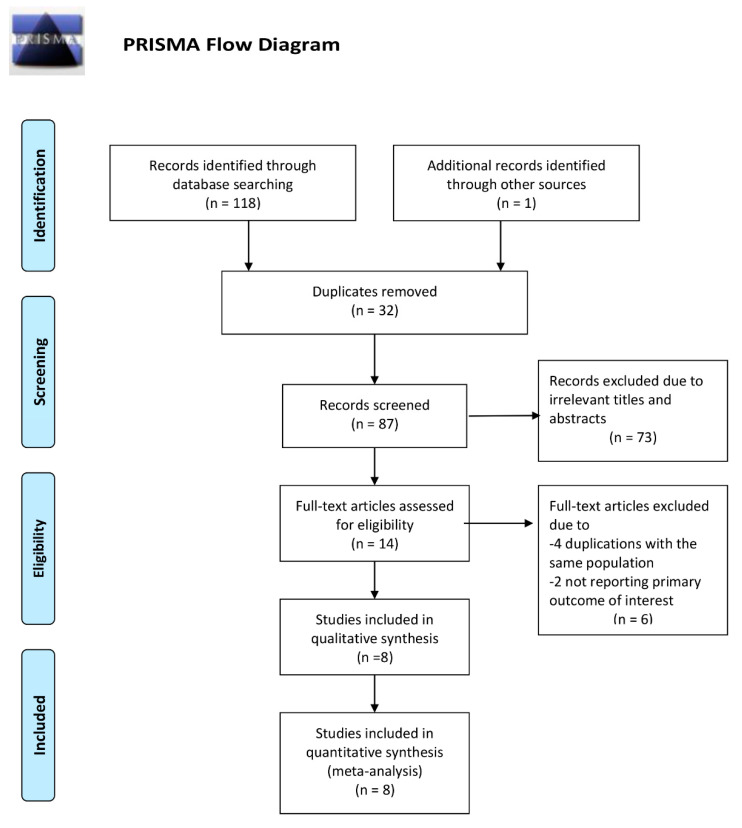
Flow diagram for study selection.

**Figure 2 medsci-08-00047-f002:**
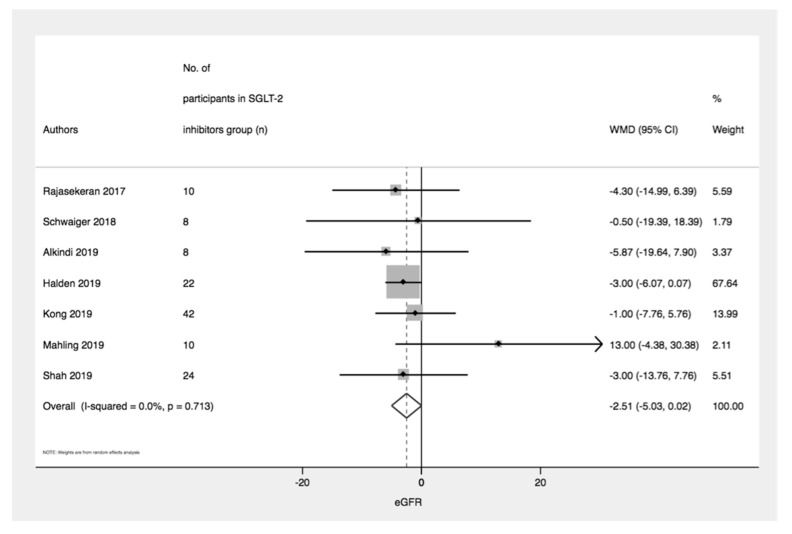
Plot displaying the pooled weighted mean difference of estimated glomerular filtration rate (eGFR), comparing levels at baseline and end of study.

**Figure 3 medsci-08-00047-f003:**
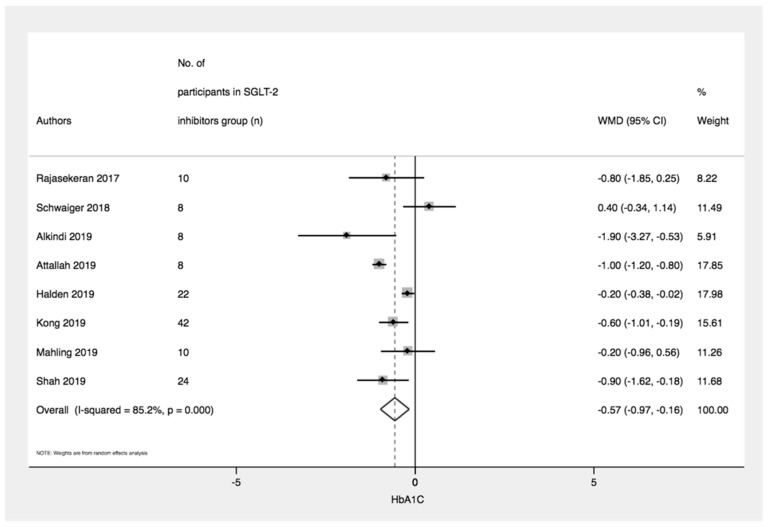
Forest plot displaying the pooled weighted mean difference of glycated hemoglobin (HbA1C), comparing levels at baseline and end of study.

**Figure 4 medsci-08-00047-f004:**
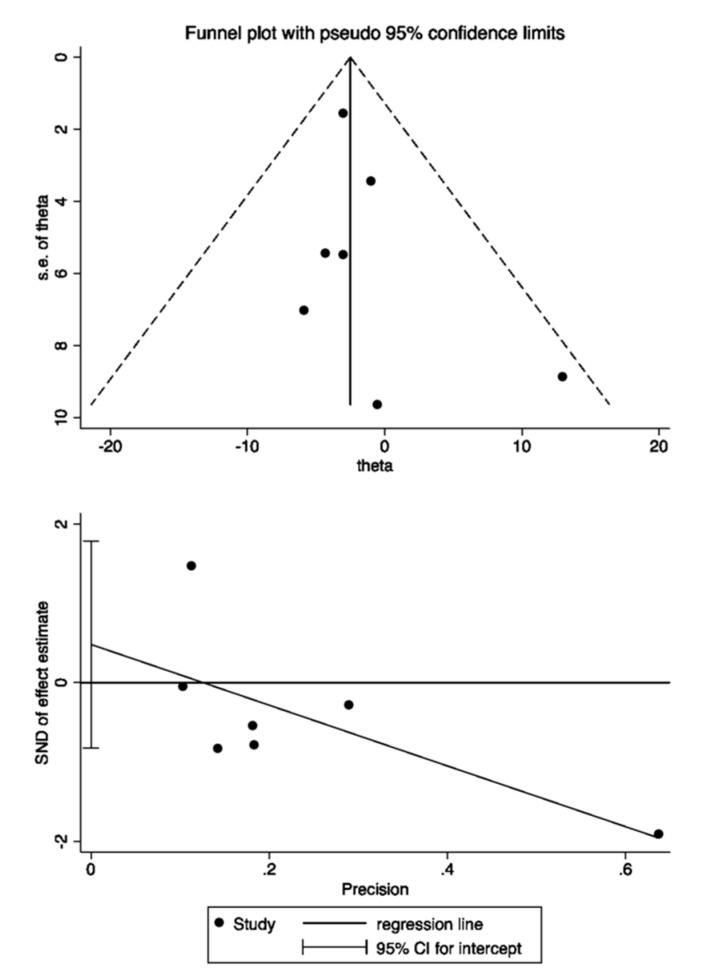
Assessment for publication bias by Funnel plot and Egger test.

**Table 1 medsci-08-00047-t001:** Characteristics of the included studies in this meta-analysis analyzing the efficacy and safety profiles of sodium-glucose co-transporter 2 (SGLT-2) inhibitors on treating kidney transplant recipients with post-transplant diabetes mellitus.

Authors	Country	Treatment Medication	Number of Participants Treated with SGLT-2 Inhibitors	Mean Age (Years)	%Male	%DM Prior to KT	F/U (Months)	GFR Baseline (mL/min/1.73 m^2^)	HbA1C	Transplant Duration (Years)	Quality Assessment
Rajasekeran 2017 [50]	Canada	Canagliflozin	10	56.7 ± 12.4	70%	80%	8	70.8 ± 18.3	8.1 ± 1.4	4.0 ± 3.4	good
Schwaiger 2018 [51]	Austria	Empagliflozin	8	56.5 ± 7.9	50%	0%	12	54.0 ± 23.8	6.7 ± 0.7	5.8 ± 4.8	good
Alkindi 2019 [52]	UAE	Empagliflozin or Dapagliflozin	8	56.8 ± 13.7	75%	25%	24	75.8 ± 13.4	9.3 ± 1.4	9.6 ± 6.4	good
Attallah 2019 [53]	UAE	Empagliflozin	8	45.9 ± 6.6	50%	50%	12	NR	8.1 ± 0.2	19.9 ± 6.0	good
Halden 2019 [43]	Norway	Empagliflozin	22	63	77%	0%	6	66 ± 10.5	6.9 ± 0.4	3	good
		Placebo	22 (control)	59	77%	0%	6	59 ± 9.5	6.8 ± 0.3	3	
Kong 2019 [56]	South Korea	Dapagliflozin	42	NR	NR	NR	12	60.3 ± 17	7.5 ± 1.1	NR	fair
Mahling 2019 [54]	Germany	Empagliflozin	10	66.0 ± 12.6	80%	30%	12	57 ± 19.3	7.3 ± 1.0	5.9 ± 3.3	good
Shah 2019 [55]	India	Canagliflozin	24	53.8 ± 7.1	96%	83%	6	86 ± 20	8.5 ± 1.5	2.7	good

NR, not reported; UAE, United Arab Emirates.

**Table 2 medsci-08-00047-t002:** Summary effects of SGLT-2 inhibitors on outcomes of interest among kidney transplant recipients with post-transplant diabetes mellitus, comparing levels at baseline and end of study.

Parameters	Number of Study	Sample Size	WMD	95% CI	*p*-Value	I^2^
eGFR	7	124	−2.51 mL/min/1.73 m^2^	(−5.03, 0.02)	0.06	0
Serum creatinine	5	58	−0.05 mg/dL	(−0.13, 0.03)	0.21	0
Urine protein-creatinine ratio	3	38	−211 mg/g	(−655, 232)	0.35	93.2
HbA1C	8	132	−0.57%	(−0.97, −0.16)	0.006	85.2
SBP	6	82	−3.24 mmHg	(−7.92, 1.45)	0.18	21.3
DBP	6	82	−1.49 mmHg	(−3.81, 0.83)	0.21	0
BMI	3	38	−1.20 kg/m^2^	(−2.67, 0.27)	0.11	21.4
Weight	8	132	−2.15 kg	(−3.07, −1.23)	<0.001	0

eGFR, estimated glomerular filtration rate; HbA1C, glycated hemoglobin; SBP, systolic blood pressure; DBP, diastolic blood pressure; BMI, body mass index; WMD, weighted mean difference; CI, confidence interval.

**Table 3 medsci-08-00047-t003:** Summary effects of SGLT-2 inhibit ors on outcomes of interest among kidney transplant recipients with post-transplant diabetes mellitus, comparing levels at baseline and 6 months, 12 months, and subgroup analysis on types of SGLT-2 inhibitors.

Parameters	Number of Study	Sample Size	WMD	95% CI	*p*-Value	I^2^
At 6 months						
eGFR	5	72	−2.38 mL/min/1.73 m^2^	(−5.13, 0.37)	0.09	0
Serum creatinine	NA					
Urine protein-creatinine ratio	3	38	−211 mg/g	(−655, 232)	0.35	93.2
HbA1C	5	72	−0.22%	(−0.62, 0.18)	0.28	41.6
SBP	4	62	−6.38 mmHg	(−15.54, 2.80)	0.17	64.3
DBP	4	62	−2.54 mmHg	(−6.67, 1.59)	0.23	47.2
BMI	3	38	−0.80 kg/m^2^	(−1.38, −0.22)	0.007	0
Weight	2	46	−2.49 kg	(−4.15, −0.84)	0.003	0
At 12 months						
eGFR	4	68	−0.35 mL/min/1.73 m^2^	(−5.66, 4.97)	0.90	0
Serum creatinine	3	24	−0.05	(−0.15, 0.05)	0.32	0
Urine protein-creatinine ratio	NA					
HbA1C	5	76	−0.58%	(−1.12, −0.05)	0.03	79.8
SBP	3	26	−7.25 mmHg	(−16.04, 1.54)	0.11	0
DBP	3	26	−5.24 mmHg	(−11.19, 0.72)	0.09	0
BMI	2	16	−2.70 kg/m^2^	(−6.03, 0.61)	0.11	16.1
Weight	5	76	−1.97 kg	(−3.21, −0.73)	0.002	0
Empagliflozin						
eGFR	3	40	0.31 mL/min/1.73 m^2^	(−8.27, 8.88)	0.94	37.5
Serum creatinine	2	16	0.02	(−0.13, 0.16)	0.84	0
Urine protein-creatinine ratio	3	38	−211 mg/g	(−655, 232)	0.35	93.2
HbA1C	4	48	−0.31%	(−0.91, 0.28)	0.30	92.9
SBP	3	40	1.55 mmHg	(−3.05, 6.15)	0.51	0
DBP	3	40	−1.14 mmHg	(−5.37, 3.09)	0.60	15.8
BMI	2	30	−0.82 kg/m^2^	(−1.41, −0.24)	0.006	0
Weight	4	48	−2.17 kg	(−3.20, −1.15)	<0.001	0
Canagliflozin						
eGFR	2	34	−3.65 mL/min/1.73 m^2^	(−11.24, 3.93)	0.35	0
Serum creatinine	2	34	−0.04	(−0.16, 0.07)	0.48	0
Urine protein-creatinine ratio	NA					
HbA1C	2	34	−0.87%	(−1.46, −0.27)	0.004	0
SBP	2	34	−7.15 mmHg	(−14.27, −0.03)	0.04	0
DBP	2	34	−2.49 mmHg	(−6.87, 1.89)	0.27	0
BMI	NA					
Weight	2	34	−2.14 kg	(−4.43, 0.16)	0.07	0

NA, not applicable.

**Table 4 medsci-08-00047-t004:** Safety profiles of SGLT-2 inhibitors among kidney transplant recipients with post-transplant diabetes mellitus.

Authors	Urinary Infection	Genital Mycosis	Euglycemic Ketoacidosis	Acute Kidney Injury	Acute Rejection	Ulcer	Cellulitis
Rajasekeran 2017	0	0	NR	0	0	NR	1
Schwaiger 2018	3	0	0	NR	0	NR	NR
Alkindi 2019	1	0	0	0	0	NR	NR
Attallah 2019	2	NR	0	0	0	NR	NR
Halden 2019	3	1	NR	NR	0	NR	NR
Kong 2019	3	NR	NR	NR	NR	NR	NR
Mahling 2019	2	NR	0	1	NR	1	NR
Shah 2019	0	0	NR	NR	NR	NR	NR
%incidence proportion	43.8%	1.4%	0%	3.6%	0%	10%	10%

NR, not reported.

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
