# Peer review of "Efficacy and Safety of SGLT-2 Inhibitors for Treatment of Diabetes Mellitus among Kidney Transplant Patients: A Systematic Review and Meta-Analysis"

_medsci, 2020, doi:10.3390/medsci8040047_

Round 1
Reviewer 1 Report
see the uploaded file

Author Response
Response to Reviewer 1’s Comments
Scientific contribution
A comprehensive search of several databases investigated efficacy and safety of SGLT-2 inhibitors in kidney transplant patients with DM. Study results were pooled and analyzed utilizing random effects model-KTr.
Response: We truly appreciate your comments
MATERIAL AND METHODS
Study methods is very clear and detailed. The Prisma flow diagram is clear and useful. It may be interesting to report data about RCT of the control group.
Response: Thank you for your suggestion. We appreciate your great input. Baseline characteristics of control group reported by Halden was added in Table 1
RESULTS
There are few reports in literature about the efficacy, the safety of sGLT2i therapy in Kidney transplant recipients with diabetes. Even if these results are lack of a control group results are interesting.
Response: We truly agree with your point. Our systematic review and meta-analysis primarily provided insight of efficacy and safety of SGLT-2 inhibitors among kidney transplant patients when satisfactory answers not yet available.
DISCUSSION
The discussion is wide and carefully detailed. In our opinion it would be suitable to stress the nephroprotective properties of sGLT2i in addition to direct intrarenal effects related to hemodynamic changes also by their anti-inflammatory / antioxidative / antifibrotic activities that could be harmful also in kidney transplant recipients.
Response: The following sentence was added in the discussion.
“In addition to the effect on diminishing intraglomerular hyperfiltration and hypertension, SGLT-2 inhibitors also demonstrate anti-inflammatory, antifibrotic and protective effects against deregulation of extracellular matrix evidenced by a reduction in serum levels of TNF receptor 1, IL-6, matrix metalloproteinase 7 and fibronectin 1 [9]. Moreover, SGLT-2 inhibitors also reduce serum level of leptin, c-reactive protein and IL-1β secretion via ROS-NLRP3-caspase-1 pathway [10,11]. Furthermore, SGLT-2 inhibitors also showed antioxidative effects via activation of SIRT1/AMPK, suppression of Akt/mTOR signaling pathway [12]. Some studies also found a decreased level in myeloperoxidase among SGLT-2 inhibitors group suggesting a lower oxidative damage to vascular endothelium [13]. These potential effects of SGLT-2 inhibitors might benefit kidney transplant recipients in which inflammation and fibrosis are the major process in allograft rejection.”
Referee’s conclusion
In summary, in our opinion, this manuscript needs a minor revision because the study is well explained, and the results are carefully presented. Limits of the revision are already expressed as you don’t have a control group so reducing the strength of the study. Although these limits we think that this manuscript may represent a beginning to larger study comparing not only clinical outcome.
Response: Thank you for your thoughtful comments.
We greatly appreciated the reviewer’s time and comments to improve our manuscript. The manuscript has been improved considerably by the suggested revisions.

Reviewer 2 Report
This paper contains valuable information regarding the safe use of SGLT2-inhibitors in transplant patients with diabetes mellitus. I have no further comments.
Author Response
We thank you for reviewing our manuscript and for your critical evaluation. We greatly appreciated the editors’ time and comments to improve our manuscript. The manuscript has been improved considerably by the suggested revisions.
Reviewer 3 Report
In the recent meta-analysis, the authors conducted an analysis of the efficacy and side effects of SGLT2 inhibitors in kidney transplant recipients.
The analysis is very important from practical point of view and deserves attention.
I do not see any methodological errors. The work is well written and I congratulate the authors on a very good idea.
Author Response
We thank you for reviewing our manuscript and for your critical evaluation.
We greatly appreciated the reviewers and editors’ time and comments to improve our manuscript. The manuscript has been improved considerably by the suggested revisions.
This manuscript is a resubmission of an earlier submission. The following is a list of the peer review reports and author responses from that submission.
Round 1
Reviewer 1 Report
This is an interesting study about the use of SGLT-2 inhibitors in patients with diabetes mellitus after renal transplantation. The available data on this topic are scarce, and consequently the meta-analysis included only 135 patients, most of them being reported in case series. The results are sound clearly presented.
The study’s uniqueness in reporting safety and efficacy data on SGLT-2 inhibitors in renal transplantation is also its main drawback. Since the available data are limited, the meta-analysis consisted probably mainly of case series (7 of the 8 papers) which are by definition not studies. Furthermore, important information on patients comorbidities, antidiabetic medication, immunosuppression regimens and complications of transplantation are missing. Lastly, other important safety issues, such as hypovolemia related adverse events are not reported (Menne J et al PloS Med 2019)
Unfortunately, the evidence provided by that study is by far not sufficient to draw reliable conclusions which could help clinicians. More data are required to achieve that.
Reviewer 2 Report
Meta-analysis by Chewcharat and colleagues examines SLGT2 inhibitors and outcomes in kidney transplant recipients with diabetes. Although the final number of patients included in the analyses is small (132 patients from 8 studies), this reflects the current knowledge in the field. Subgroup analysis was further available for empagliflozin and canagliflozin.
I have no major critiques. There are minor grammatical issues that should be corrected prior to publication:
Page 4: "One study was randomized controlled trials" should be restated as "One study was a randomized controlled trial"
Page 7: "There were no significant decline in eGFR" should be restated as "There was no significant change in eGFR"
Page 10: "There were no significant change in kidney functions" should be restated as "There were no significant changes in kidney function..."
Later in same paragraph, "kidney functions" should be corrected to "kidney function".
Page 12: "data on allograft kidney biopsy are limited and required future studies" should be restarted as "data on allograft kidney biopsy are limited and requires further study".
In the following paragraph, "...including cardiovascular events, delay graft rejection, lower all-cause mortality, as well as safety profiles" should be reworded and shortened to "including cardiovascular events, delayed graft rejection and mortality risk."
Reviewer 3 Report
General remark: As mentioned in the limitation section, the meta-analysis is based on only one randomized controlled trial and 7 case series, without control group and with a limited number of patients. There was no adjustment of potential confounders. This are major issues if you want to investigate the safety and efficacy of SGLT-2 inhibitors in kidney transplant patients. Sometimes, it is just too early to perform a meta-analysis on a subject. Page 1 Abstract: - When you use an abbreviation, you should give first the full name. - “SGLT-2 inhibitors demonstrated a significantly lower HbA1C …” I don’t think medication has a lower HbA1c: e.g. patients treated with SGLT-2 inhibitors. Page 4 The numbers used in the PRISMA 2009 Flow diagram are to my opinion not correct. The paper should be rewritten by a native English speaker as several sentences are grammatically incorrect: e.g. “One study was randomized controlled trials,(30)”